# SHINRA: Structuring Wikipedia by Collaborative Contribution

**Satoshi Sekine**                                   SATOSHI.SEKINE@RIKEN.JP
**Akio Kobayashi**                                   AKIO.KOBAYASHI@RIKEN.JP
**Kouta Nakayama**                                   KOUTA.NAKAYAMA@RIKEN.JP
*1-4-1 Nihonbashi, Chuo-ku, 15th floor*
*Tokyo, 103-0027, Japan*

## Abstract

This paper describes a Knowledge Base construction project, SHINRA. It is a project to structure Wikipedia knowledge with "Resource by Collaborative Contribution (RbCC)" scheme utilizing Automatic Knowledge Base Construction (AKBC) systems. The ultimate goal of the project is to create a huge and well-structured knowledge base which is essential for many NLP applications, such as QA, Dialogue systems and explainable NLP systems.

Based on "Resource by Collaborative Contribution (RbCC)" scheme, we conducted a shared task for structuring Wikipedia for the purpose of attracting participants, but at the same, submitted results are used to construct a knowledge base. There have been a lot of shared-tasks, but the results of the participated systems are not well used. These results, which we believe are the great resources, are abandoned after the evaluation is over. We conducted a project which is a shared-task on AKBC, but at the same time the results of the systems are gathered and used to produce even the better results than the best participated system by ensemble learning. One of the trick is that we, the organizers, don't notify which is the test data to the participants. So they have to run their systems on all entities in Wikpedia and submit them, even though the evaluation results are reported only on 100 test data among the entire data. By this methods, the organizers will receive the structured information for all entities from the participants. The accumulated results are later become open to the public, and will be used to build the even better structured knowledge by the ensemble learning methods, for example. In other words, this is a project to use AKBC systems as a tool to construct a huge and well-structured Knowledge Base in collaborative manner.

We will report "SHINRA2018" project which runs under the RbCC scheme. The task is the attribute extraction task, i.e. to extract the values of the attributes from Wikipedia pages. We have categorized most of Japanese Wikipedia entities (namely 730 thousand entities) into the 200 Extended Named Entity (ENE) categories prior to this project. Based on this data, the shared-task is to extract the values of the attributes which are defined for each category from texts and infobox in the Wikipedia pages. We gave out the 600 training data and the participants are required to submit the attribute-values for all remaining entities of the same category. Then 100 data, which is hidden to the participants for each category are used to evaluate the system as a shared-task.

To this project, 8 groups submitted the results based on 15 systems. We conducted a preliminary ensemble learning on the outputs in order to demonstrate how the RbCC scheme works. The results of the ensemble learning shows a huge improvement over the best single system for each category. The improvements are 15 F-score on a category of "airport" in which the best system achieves 72 F-score, and 9 F-score on the average. This results show that the RbCC scheme is very effective.

Based on this results, we decided to conduct three tasks in 2019; multi-lingual categorization task (ML), extraction for the same 5 categories in Japanese with a larger training data (JP-5) and extraction for 30 new categories in Japanese (JP-30).

## 1. Introduction

Wikipedia is a great resource as a knowledge base of the entities in the world. However, Wikipedia is created for human to read rather than machines to process. Our goal is to transform the current Wikipedia to a machine readable format based on a clean structure. There are several machine readable knowledge bases (KB) such as CYC[Lenat, 1995], DBpedia[Lehmann et al., 2015], YAGO[Mahdisoltani et al., 2015], Freebase[Bollacker et al., 2008], Wikidata[Vrandečić and Krötzsch, 2014] and so on, but each of them has problems to be solved. CYC has a coverage problem, and others have a coherence problem due to the fact that these are based on Wikipedia and/or created by many but inherently incoherent crowd workers. In order to solve these problems, we started a project for structuring Wikipedia using automatic knowledge base construction (AKBC) shared-task using a cleaner ontology definition.

The automatic knowledge base construction shared-tasks have been popular for decades. In particular, there are popular shared-tasks in the field of Information Extraction, Knowledge Base population and attribute extraction, such as KBP[U.S. National Institute of Standards and Technology (NIST) , 2018] and CoNLL. However, most of these tasks are designed only to compare the performances of participated systems, and to find which system ranks the best on limited test data. The outputs of the participated systems are not shared and the results and the systems may be abandoned once the evaluation task is over.

We believe this situation can be improved by the following changes:

1. designing the shared-task to construct knowledge base rather than only evaluating on limited test data

2. making the outputs of all the systems open to public so that anyone can run ensemble learning to create the better results than the best single system

3. repeating the task so that we can run the task with the larger and better training data from the output of the previous task (active learning and bootstrapping)

We conducted "SHINRA2018" with the aforementioned ideas, we call it "Resource by Collaborative Contribution (RbCC)". In this paper we report the first results and the future directions of the project.

The task is to extract the values of the pre-defined attributes from Wikipedia entity pages. We used Extended Named Entity (ENE) as the definition of the category (in total 200 categories in the ontology) and the attributes (average of 20 attributes) for each category. We have categorized most of the entities in Japanese Wikipedia (namely 730 thousand entities) into the ENE categories prior to this project. Based on this data, the shared-task is to extract values of the attributes defined for the category of each entity. At the SHINRA2018 project, we limited the target categories to 5, namely, person, company, city, airport and chemical compound. We gave out the 600 training data each for 5 categories at and the participants are supposed to submit the attribute-values for all remaining entities

of the categories in Japanese Wikipedia. Then 100 data out of the entire pages of the category are used at the evaluation of the participated systems in the shared-task. For example, there are about 200K person entities in Japanese Wikipedia, and the participants have to extract the attribute-values, such as "birthday", "the organizations he/she have belonged", "mentor" and "awards" from all the remaining entities (i.e. 199.4K = 200K-600 entities). Before starting the project, the participants signed the contract that all the output will be shared among all participants, so that anyone can conduct the ensemble learning on those outputs, and hence create a better knowledge base than the best system in the task. Note that, for the sake of participant's interest, i.e. a company may want to keep the system as their property, the outputs are required to be shared, but their systems are not necessarily to be shared. A promising results of the ensemble learning is achieved and we envision that it will lead to the cleaner machine readable knowledge base construction.

## 2. Related Work

Structured knowledge bases have considered as one of the most important knowledge resources in the fields of Natural Language Processing. There are several major projects targeted to construct structured knowledge bases in the past. One of the earliest project is CYC, and more recently there are Wikipedia based projects such as DBpedia, Yago, Freebase and Wikidata. Moreover, there are some shared-tasks aiming to build techniques for knowledge base structuring such as KBP and CoNLL. We will introduce these resources and projects and describe the points we consider as issues to be solved in those projects.

CYC ontology is a large knowledge base constructed as common sense knowledge[Lenat, 1995]. This is one of the large projects in the AI in 80-90's, which use the human labor to construct knowledge base. The cost of construction and maintenance of the handmade knowledge bases for the general domain is very high, and it is known that the knowledge bases have problems in the coverage and the consistency.

DBpedia is a more recent project to construct a structured information from the semi-structured data in Wikipedia such as infoboxes or categories[Lehmann et al., 2015]. DBpedia also has a problem of accuracy, coverage, and coherence. Like CYC, it is also created by human, but in this case, those who worked on creating the knowledge are non-experts of ontology. For example, "Shinjuku Station", which is a railway station, has a category "Odakyu Electric Railway", which is a railway company using the station. Of course, a station can't be an instance of a railway company, so this is not appropriate category. There are so many examples like this in DBpedia. Also there are many inconsistencies in the category hierarchy, and the attributes defined in the KB are not well organized in many categories.

Yet Another Greater Ontology (YAGO) is a ontology constructed by mapping Wikipedia articles to the WordNet synsets[Mahdisoltani et al., 2015]. YAGO has adopted attributes information extracted from infoboxes like as DBpedia because no attributes defined in WordNet synsets.

Freebase is a project to construct a structured knowledge base by crowdsourcing, same as Wikipedia[Bollacker et al., 2008]. However, by the crowdsource approach, Freebase doesn't have a well-organized ontology. It has many noises and lack of coherence because these

were created by unorganized crowds. Currently, the project of Freebase has paused and integrated into Wikidata.

Wikidata is aiming to be a structured knowledge base based on corwdsourcing scheme. [Vrandečić and Krötzsch, 2014]. Wikidata also have noises and lack of coherence because it has constructed by bottom-up approach same as Wikipedia and Freebase. For example, just comparing the definition of "city", "town" and "human settlements", we can easily observe inconsistencies in the property (the number of properties are 30, 0 and 6, respectively), there are very biased properties such as "Danish urban area code" in "human settlements", there are many related ambiguous entities, such as "like a city", "city/town" and so on. Also, the category inconsistency can be easily found, for example, "city museum", "mayor" are subcategory of "city", although a mayor is not an instance of "city". Wikipedia allows topics to be included in a category, however, this policy prevents to make the category hierarchy as a well-designed ontology.

KBP is a shared task organized by NIST for establishing a technology to construct a structured knowledge base from non-structured documents[U.S. National Institute of Standards and Technology (NIST) , 2018]. KBP mainly consists of two tasks. One task is an Entity Discovery and Linking (EDL) which is to find and identify an entity defined in DB from documents. Another one is a Slot Filling which is to extract attribute information of the entity. KBP in general is limited entity types to Person, Location, and Organization in contrast to Wikipedia's wide coverage, and mostly it is a competition based project and no resource creation purpose.

FIne Grained Entity Recognition (FIGER) is a project to identify 112 types of named entity classes that are finely defined, such as ENE, from documents[Ling and Weld, 2012]. The category of FIGER seems a bit biased, and it doesn't have attribute definitions for each category.

## 3. Extended Named Entity

In order to create structured knowledge base which is useful for NLP applications, we have learned that well structured ontology is needed and it has to be designed top-down manner. DBpedia, Freebase and Wikidata are created by crowds in bottom up manner,and these have inconsistent entries, imbalanced ontologies and adhoc attributes, as we described in the previous section. We believe the major cause is the fact that these are created bottom-up manner, and a top-down design is essential to design the ontology and the attributes. As a top-down designed ontology for named entities, we employed the "Extended Named Entity (ENE) hierarchy" in our project. Extended Named Entity (ENE) is a named entity classification hierarchy along with the attribute definition for each category [Sekine, 2008, Sekine et al., 2002]. It includes fine-grained 200 categories of entities in hierarchy of up to 4 layers. It contains not only the finer categories of the typical NE categories, such as "city" and "lake for "location" or "company" for "organization", but also contains new named entity types such as "products", "event", "position" and so on. These categories are designed to cover a large amount of entities in the world using encyclopedia and many other resources. Figure 1 shows the ENE definition, version 7.1.0. Attributes are defined based on the investigation of the entities in each category. For example, the attributes for "airport" categories are as follows: "Reading", "IATA code", "ICAO code", "nickname",

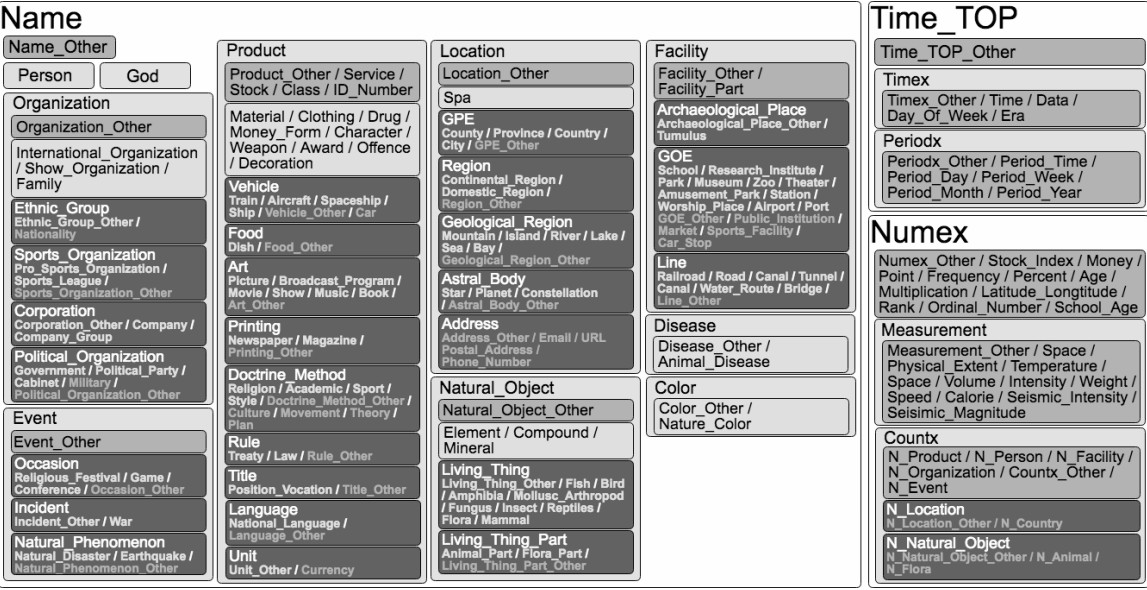

Figure 1: Definition of Extended Named Entity Hierarchy

"name origin", "number of users per year", "the year of the statistic", "the number of airplane landing per year", "longitude", "latitude", "location", "old name", "elevation", "big city near by", "number of runaway" and so on. Please refer to the HP for the complete definition.

## 4. Categorization of Wikipedia Entities

In order to conduct the shared-task of the attribute-value extraction on Wikipedia entities, first, we have to assign one or more categories to each Wikipedia entity. For example, we have to know the Wikipedia page of "Chicago O'Hare Airport" belongs to an airport entity, and we supposed to extract attribute-value of airport from the page. We have annotated one or more of 200 ENE categories to 782,406 entities of Japanese Wikipedia (201711 version) prior to this project. At the annotation, we have excluded the less popular entities which have less than 5 incoming links (151K entities), and non-entity pages (about 53K pages) such as common nouns and simple numbers. This annotation was done by Machine learning method followed by hand checking on less reliable ones. The machine learning [Masatoshi et al., 2018] was conducted with 20K training data created by hand. Then a human check was conducted on the machine learning outputs with less reliable scores. In order to see the quality of the categorization, we evaluate a sample data by multiple annotators to see the accuracy of the data and we observed the accuracy of the categorization is 98.5%. The remaining 1.5% are those which are ambiguous in nature and are very difficult even for the human annotators. Table 1 show the most frequent categories with the frequencies in the data.

Table 1: 14 Most Frequent Categories

| Category | Frequency |
|---|---|
| Person | 247,983 |
| City | 45,306 |
| Music | 41,049 |
| Artifact other | 33,453 |
| Broadcast program | 32,050 |
| Company | 26,746 |
| School | 23,609 |
| Literature | 18,515 |
| Movie | 17,901 |
| Train station | 16,901 |
| Sports event | 15,863 |
| Road | 15,360 |
| Method other | 14,766 |
| Play group | 10,181 |

## 5. Shared-Task Definition

In this section, we will describe the definition of the shared-task. The task is to extract the attribute-values of entities from the Wikipedia page. For this year's task (SHINRA2018), we picked 5 categories, namely "person", "city", "company", "airport" and "chemical compound" for the shared-task. This selection was done on the largest subcategories of "person", "location", "organization", which are the traditional three categories of named entity (person has no subcategories, and itself is the only category in ENE) "Airport" is selected as a category which has well-structured infobox in Wikipedia, on the other hand, "chemical compound" is selected because the information in infobox is not quite satisfactory for NLP purpose. The infobox for "Chemical compound" contains the factual information such as "boiling temperature", or "chemical formula", but it doesn't contain information such as "usage" or "production method", which we believe are important attributes, for example, for QA purpose.

We gave out 600 training data for each category. In the training data, all attribute-values mentioned in the Wikipedia page are manually extracted and form the training data in JSON format. The participants also received the list of all entities, i.e. Wikipedia pages, for 5 categories, and they are required to extract attribute-values from Wikipedia pages of all entities. The evaluation is conducted on 100 entities for each category, but the participants are not notified which 100 entities are used for evaluation even after the evaluation is over. This is for the purpose of the data construction so that the participants have to do their best to produce the output for all data, and the purpose of the future comparison (if the test data is known, the participants could tune their system to the test data even unintentionally through a number of experiments). The results are reported by precision, recall and F1 scores, as usual. The systems submitted before the deadline are reported as the formal results and the results submitted after the deadline are reported as

a reference results. In the ensemble learning experiment which will be reported later in this paper, we use all the results regardless of the formal or reference results so that we can achieve the best results for the resource construction purpose. The participants are not required to submit the results for all 5 categories, as some participants might be interested in a particular category or may have smaller machine resources to run their system for all the categories.

## 6. Creating the Training and Test Data

The manual creation of the training and test data was not easy. In this section, we briefly described the data preparation, as this process is very interesting and could be a topic of one another paper by itself. We tried to use three types of annotators to create the data as a preliminary experiment.

- Experts of the construction of linguistic data

- Students who are supervised by experts

- Workers on the crowdsourcing (Lancers)

As we can imagine intuitively, we found out that the upper in the list, the more expensive, but at the same time the more accurate. Also, we found that the crowdsourcing has relatively high coverage based on our strategy of the crowdsourcing. The task of crowdsourcing is designed with three stages. The first stage is to identify the sections where the given attribute-value is written. In this stage, even the worker find the value in the page, they are not requested to extract the value. This identification of the sections will be repeated until two workers found no value is found, because some attributes have multiple values in one page. Then in the second stage, the values are extracted from the sections which are identified to contain the value(s). The final stage is to check if the extracted value is really the value for the attribute. Maybe this careful strategy might lead to produce the relatively high coverage. Based on the preliminary annotation experiments, we decided to use "expert" and "crowd" at the final data creation. The first round annotation is done by both "expert" and "crowd" independently for the same attributes, and then both results are merged to create the final annotation by another "expert" (different from the one who annotate it initialy). The inter-annotator agreements between "crowd" and "experts" are 60-80% and that between "experts" are 80-90% depending on the attributes. The coverage by "crowds" is relatively high and it suggest missing information by the first "expert" at the final annotation by the "expert".

## 7. SHINRA2018 Shared-Task: Results

In this section, we will report the results of the shared-task. Five months are given to the participants to develop their systems and run their experiments from April to September 2018. 16 systems by 8 participants are submitted at SHINRA2018. The first two columns in Table 2 shows the participants (some in abbreviations) and their methods. Here "pattern" means that they created a hand-made patterns for the attribute-value extraction, and "DL" means some sort of "Deep Learning". "DrQA" is an open source QA system adapted

Table 2: SHINRA2018 Results

| Participants | method | ENE Category | | | | |
|---|---|---|---|---|---|---|
| | | Person | Company | City | Airport | Compound |
| TUT | Pattern | 0.20 | 0.41 | 0.28 | **0.72** | |
| OPU | Pattern | 0.19 | | | | |
| | Pattern + Heuristics | 0.16 | | | | |
| NUT | Pattern+LightGBM | | | | | 0.42 |
| Sansan | Pattern | | 0.30 | | | |
| Fuji Xerox | NCRFpp | 0.31 | | | 0.38 | 0.15 |
| | RDFDNN | | | | | 0.15 |
| | NCRFPP | | 0.30 | 0.43 | 0.42 | 0.39 |
| | RDFDNN | | 0.28 | | 040 | 0.37 |
| TOPPAN | BRNN/LSTM | | 0.29 | | 0.35 | |
| | Pattern | | 0.33 | | | |
| | BRNN/LSTM + Pattern | | 0.34 | | 0.42 | |
| | Pattern | | 0.41 | | | |
| Unisys | DrQA | | **0.53** | | 0.67 | **0.47** |
| | DrQA | **0.44** | | 0.42 | | |
| AIP | RNN | 0.36 | 0.38 | **0.46** | 0.71 | 0.46 |

Japanese QA system. In this system, the participant transformed the infobox into a sentence by pattern, e.g. "The birthday of Barack Obama is August 4, 1961" and attribute-values to be extracted is transformed to a question, e.g. "Who is the father of Barack Obama?" in order to extract "father" attribute-value of "Barack Obama" entity. Then they train and run DrQA for all categories together. For RbCC purpose, it is quite valuable to have technologies of wide variety used in this shared-task.

The results are shown in 3rd to 7th columns in the same table. The top result is shown in bold for each category. The Unisys's DrQA system performs the best in three categories, most of which don't have so much information in infobox. As their method handles all the attributes in a single system (regardless of infobox or in the explanation sentence), the amount of training data for the system becomes relatively larger and it may receive the benefit in training data size at the situation where the training data is relatively small. TUT's pattern based system performed very well on airport category, in which the most of the required information are described in infobox, and practically only one infobox template is used in the category. Note that the category "person" has many different infobox templates depending on the vocation of the person, and the company's infobox templates vary depending on the type of the company.

## 8. Preliminary Results of Ensemble Learning

The goal of "Resource by Collaborative Contribution (RbCC)" is to produce more accurate KB than the KB created by the best single system. In order to see if RbCC scheme is practical and promising, we conducted a preliminary experiment of ensemble learning on the all system's outputs. Note that the ensemble learning is conducted in order to show an evidence if RbCC scheme is practical and promising. In the past, the ensemble learning methods have been studied with various ideas; such as Bagging [Leo Breiman and Eiman, 1994], Boosting [Freund and Schapire, 1997] or stacking [Wolpert, 1992], [Breiman, 1996], [Smyth and Wolpert, 1998]. These methods are generally used to create a high accuracy system combining more than one ML systems. However, in our situation, the outputs of many systems are given and the objective is to produce the best output out of the system outputs by ensemble learning method. Because of this, the stacking method is best suitable for our purpose, but , first, we tried two simpler methods, i.e. the simple voting method and the weighted voting method base on the accuracy on held-out data as a preliminary ensemble learning experiment.

First, we will explain the simple voting method. Assume there are $n$ systems which outputs value $v$ for an attribute of an entity. Then the value $v$ receives score $n$. Separately, we compute the threshold $t$ by maximizing the accuracy on the held-out data. If $n > t$, then we take the value $v$ as the output of the ensemble system. In practice, we split the actual test data, which contains 100 samples, into two halves; one for the held-out data and the other for the test data for this experiment. We conducted the same experiment replacing the held-out data and test data; i.e. we conducted a 2 fold cross-validation experiment on 100 test data. The other method is the weighted ensemble method. We weighted the vote of the system by the accuracy of the system on the held-out data. Instead of a sum of the number of the systems which produce attribute-value $v$, we compute the sum of the accuracy as the score for the value $v$. The way to define the threshold and the cross-validation mechanism are the same to those of the simple voting method.

We will show the precision, recall and F1 score of the baseline and the ensemble methods in Table 8. Also the relative improvement of those two methods compared to the baseline method is shown in Figure 2. The baseline method is constructed by combining the best system outputs for each category, i.e. TUT system for "airport", AIP system for "city" and Unisys system for the rest, which is better than a single system, e.g. Unisys, though. We can observe from the table and the graph that the two voting methods performs better than the baseline methods in F1 score. Also the weighted voting method performs better than the simple voting method. The improvement exceed 15 F-score on "airport" category and 4 F-score for all categories. The average improvement is 10.4 F-score. The results show the effectiveness of the ensemble learning methods and are the evidence that "Resource by Collaborative Contribution" scheme is promising and encouraging.

## 9. Future Projects: SHINRA2019

Based on the success of the SHINRA2018 project, we decided to continue this project as SHINRA2019. We are planning to conduct three tasks as follows:

- ML: Multi-lingual categorization task

Table 3: Results of Ensemble Learning

| Category | Baseline method | | | Simple voting | | | Weighted Voting | | |
|---|---|---|---|---|---|---|---|---|---|
| | Prec. | Recall | F1 | Prec. | Recall | F1 | Prec. | Recall | F1 |
| airport | 0.790 | 0.658 | 0.718 | 0.802 | 0.854 | 0.828 | **0.883** | **0.861** | **0.872** |
| city | 0.378 | **0.588** | 0.460 | 0.541 | 0.544 | 0.542 | **0.570** | 0.575 | **0.573** |
| company | **0.748** | 0.441 | 0.555 | 0.569 | 0.479 | 0.520 | 0.675 | **0.529** | **0.593** |
| person | **0.563** | 0.362 | 0.440 | 0.437 | 0.474 | 0.455 | 0.501 | **0.459** | **0.479** |
| compound | **0.750** | 0.370 | 0.496 | 0.567 | 0.616 | 0.590 | 0.674 | **0.653** | **0.663** |
| average | 0.574 | 0.479 | 0.522 | 0.575 | 0.579 | 0.577 | **0.651** | **0.602** | **0.626** |

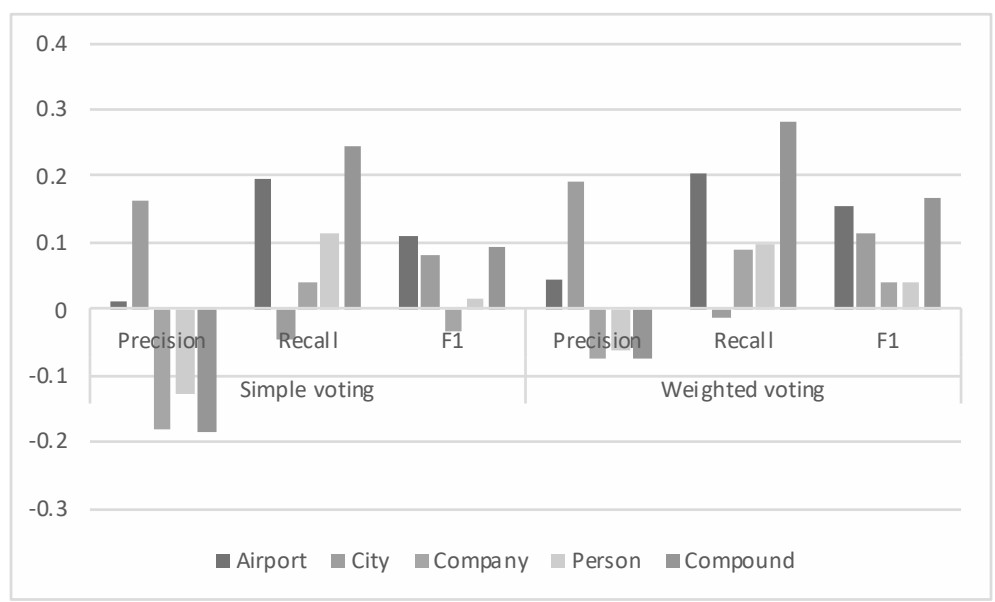

Figure 2: Relative improvements of the ensemble methods to the baseline method for each category

- JP-5: Structuring task for the same 5 categories with larger training data in Japanese

- JP-30: Structuring task for 30 new categories with 100 training data in Japanese

The multi-lingual task is to expand the benefit of RbCC to the knowledge base resources in languages other than Japanese. We are planning to run it on 9 languages with the largest numbers of "active users"; namely English, Spanish, French, German, Chinese, Russian, Portuguese, Italian and Arabic [Wikipedia, 2019]. Actually, Japanese is the 10th ranked language on the measure, so Wikipedias of these 9 language have more users than that of Japanese. As we don't have the category information for those 9 language Wikipedia entities, the first task is to categorize the entities. For Japanese, we annotated 20K entities as the training data for the categorization, but now we have most of the Japanese entities categorized, we can utilize this information. There are links between equivalent Wikipedia

entities in different languages. For example, we observed there are about 500K entities links from Japanese to English among 720K entities already categorized in Japanese. We can use them as the training data to categorize English entities. Likewise there are language links to other 8 language Wikipedias from Japanese Wikipedia, although the number of linked entities are much smaller and some noise may exist, the participants can use much bigger training data than that for the initial Japanese categorization experiment. As there are links between the Wikipedias of other languages and possibly different types of infoboxes exist in other language, too, the participants have a lot of information to be used in the categorization task.

JP-5 is the task to extract the attribute-values for the same 5 categories in SHINRA2018; namely "person", "company", "city", "airport" and "chemical compound". At SHINRA2018, the values are prepared without contexts. In other words even there are more than one mention of a particular attribute, we didn't give out which one is the mention to that value. For example, assume the nationality for a person is "Japan", but the same string may be mentioned in the same person page but not necessarily be meant to indicate the nationality of the person, e.g. "He left Japan", we had no means to know that the context is not for the nationality. It is similar to the situation of the distant-supervision, so it is difficult to extract only the context of nationality. At SHINRA2019, we will annotate the attribute-value in the text, so that the exact context for the value can be extracted. We are also planning to expand the size of the training data from 600 to 1500, at least for the categories "person", "company" and "city" using the output of the ensemble system. This forms a bootstrapping scheme as the project year by year.

JP-30 is the task to extract attribute-value for 30 new categories. As we mentioned in the previous section, creating the data is laborious, the size of the training data will be very small, namely 100. However the categories to be tested will be very close; 7 subcategories of Geographical Political Entities (GPE) such as country, prefecture/state and county, 8 subcategories of terrain such as mountain, island, river, lake and ocean, and organizational entities such as international organization, political organization, ethnic group and nationality. Although the number of training data is much smaller, we chose the very similar types and the similar attributes may exists. Some techniques of machine learning with adaptation might help creating a good result. We expect to build a larger training data using bootstrapping scheme, just like JP-5 at SHINRA2018 and SHINRA2019.

We hope to have many participants so that the better results can be achieved by the ensemble learning methods to all three tasks.

## 10. Conclusion

We proposed a scheme of knowledge base creation: "Resource by Collaborative Contribution". We conducted the Japanese Wikipedia structuring project, SHINRA2018, based on that scheme. Based on Extended Named Entity, the top-down definition of categories and attributed for named entities, the task is to extract the attribute-values from Japanese Wikipedia pages. 8 groups participated to the task, and the ensemble learning results shows that the RbCC scheme is practical and promising. A quite big improvement over the the best single system was achieved on "airport" category (more than 15 F-score), and the average of 8 F-score improvement was achieved using the weighted voting methods. We are

planning to conduct SHINRA2019 based on the RbCC scheme on 3 tasks. These are the multi-lingual categorization, the extraction of attribute-value on the same 5 categories, and the extraction of attribute-values on 30 new categories in Japanese.

We'd like to express our deep appreciation to all the participants and collaborators who helped this project. Without the participation, we couldn't even try the ensemble learning and achieve the goal. We are hoping to expand and spread the idea of RbCC scheme, not only limited to this kind of task and resource.

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
