# OpenReview forum: "SHINRA: Structuring Wikipedia by Collaborative Contribution"
_AKBC.ws/2019/Conference — AKBC 2019_

### Official Review · AnonReviewer2 · 2018-12-29
**The paper reports a summary of the SHINRA project for structuring Wikipedia collaborative construction scheme.**

**Rating:** 6
**Confidence:** 3

**Review:**

The paper described SHINRA2018 task that construct knowledge base rather than evaluating only limited test data. The paper repeat the task with larger and better training data from the output of the previous task.

The paper is well written in general, though there are some redundancies between abstract and introduction with exactly the same content. The SHINRA share task provide a good resource and platform for evaluating knowledge graph construction task on Japanese Wikipedia.

One of the concerns is that the paper did not really solve the first statement in abstract that it still evaluates on limited test data with 100 samples. The main contribution of the paper seems to be ensemble learning which has been proved efficient in many previous work.

---

### Official Review · AnonReviewer1 · 2019-01-09
**A practical work but lacks methodological contribution**

**Rating:** 4
**Confidence:** 4

**Review:**

This paper introduces a project for structuring Wikipedia by aggregating the outputs from different systems through ensemble learning. It presents a case study of entity and attribute extraction from Japanese Wikipedia.

My major concern is the lack of methodological contribution.
- Ensemble learning, which seems most like the methodological contribution, is applied in a straightforward way. The finding that ensemble learning gives better results than individual learners is trivial.
- Authors state that a key feature of the project is using bootstrapping or active learning. This, however, is not explained in the paper nor supported by experimental results.

Clarification or details are needed for steps introduced by section 3-6:
- In "Extended Named Entity", why would the top-down ontology ENE better than the inferred or crowd created ones? I think each of them has pros and cons.
- In "Categorization of Wikipedia Entities", is training data created by multiple annotators? what is the agreement between the multiple annotators for the test (and the training) data? How much error of the machine learning model is caused by incorrect human annotations?
- In "Share-Task Definition", "We give out 600 training data for each category." does it mean 600 entities?
- In "Building the Data", what is the performance of experts and crowds in the different stages?

Writing should be improved. Some examples:
- what does it mean by "15 F1 score improvement on a category".
- a lot of text in the abstract is repeated in the introduction.
- "For example, ”Shinjuku Station” is a kind of railway station is a type of ...": not a sentence.
- "4 show the most frequent categories": should be Table 1.
- page 8, "n ¿ t"L corrupted symbol.

As the last comment, I wonder how (much) this ensemble learning method can be better than crowd based KBC methods, as motivated by abstract and introduction. I would assume that machine learning has similar reliability issue as crowdsourcing even when ensemble learning is used.

---

### Official Review · AnonReviewer3 · 2019-01-13
**The paper described a information extraction task, but too many questions are unanswered**

**Rating:** 4
**Confidence:** 3

**Review:**

The paper tackles an important problem: extraction of structured data from unstructured text, but lack of comparison with existing approaches.

Section 1
Wikipedia is not only a knowledge based of the names in the world. Maybe the authors wanted to say the "entities of the world"?
The motivation of the paper is limited: what is the goal of the structured knowledge base? I the goal is better consistency, how to we improve consistency. There is nothing in the paper that indicate that the consistency is better than, say, manually created data with a voting mechanism. Why this approach to KB structuring is inherently coherent?


Section 2
If the only problem of CYC is the coverage, why the authors did not try to improve the coverage of CYC instead of inventing a new method?

Section 3
Why top-down design of ontology is needed. If the authors have learned it, what are the supporting evidence for it?

Section 4
No annotation reliability (e.g. the inter-annotator agreement score).

Section 5
Why "chemical compound" was selected and not "movie" or "building" as more common sub-categories?
600 data points in quite small compared to standard datasets. What was the cost of the annotations?

Section 6
Why the Workers (Lancers) were not used? What was their accuracy/cost ? Maybe the cost could compensate the low accuracy.

Section 7
Why is it scientifically interesting to know that the authors are happy ?

Section 8
Why did the authors participate in the shared task?
What are the references for stacking?
As far as I know, stacking performs poorly compared to proper inference techniques, such as CRF. Why is it different in this case?

Overall: the English writing is very approximate. I'm not a native speaker myself, but I would suggest the authors to send the paper to a native English speaker for correction.

---

### Meta-Review · Area_Chair1 · 2019-02-12
**Interesting topic but still not mature presentation**

**Recommendation:** Accept (Poster)
**Confidence:** 3

**Metareview:**

As it is clear from the reviewers comments, and also the rebuttal responses, there are still significant amount of points to improve in the paper. However, I believe it is going to be an interesting poster presentation

---

### Decision · Program_Chairs · 2019-02-15
**AKBC 2019 Conference Decision**

Accept